# microRNA-22 Inhibition Stimulates Mitochondrial Homeostasis and Intracellular Degradation Pathways to Prevent Muscle Wasting

**DOI:** 10.3390/ijms26209900

**Published:** 2025-10-11

**Authors:** Simone Tomasini, Emanuele Monteleone, Anna Altieri, Francesco Margiotta, Fereshteh Dardmeh, Hiva Alipour, Anja Holm, Sakari Kauppinen, Riccardo Panella

**Affiliations:** 1Center for RNA Medicine, Department of Clinical Medicine, Aalborg University, 2450 Copenhagen, Denmark; ska@dcm.aau.dk; 2Department of Life Sciences and Systems Biology, University of Turin, 10124 Torino, Italy; emanuele.monteleone@unito.it; 3Resalis Therapeutics Srl, Via E. De Sonnaz 19, 10121 Torino, Italy; anna.altieri@resalistherapeutics.com; 4Department of Neuroscience, Psychology, Drug Research and Child Health (NEUROFARBA), Pharmacology and Toxicology Section, University of Florence, Viale G. Pieraccini 6, 50139 Firenze, Italy; francesco.margiotta@unifi.it; 5Department of Health Science and Technology, Aalborg University, 9220 Aalborg, Denmark; feda@hst.aau.dk (F.D.); hiva@hst.aau.dk (H.A.); 6RNA Therapeutics, Translational Research Centre, Rigshospitalet, 2600 Glostrup, Denmark; anja.holm.nordvang@regionh.dk; 7Cell and Drug Technologies, Department of Health Technology, Technical University of Denmark, 2800 Kongens Lyngby, Denmark; 8European Biomedical Research Institute of Salerno (EBRIS), Via Salvatore de Renzi 50, 84125 Salerno, Italy

**Keywords:** microRNA, miR-22, oxidative metabolism, muscle atrophy, DMD, autophagy, fibrosis, miRNA therapeutics, ASO, antimiRs

## Abstract

MicroRNA-22 (miR-22) is a negative regulator of mitochondrial biogenesis, as well as lipid and glucose metabolism, in metabolically active tissues. Silencing miR-22 holds promise as a potential treatment of obesity and metabolic syndrome, as it restores metabolic capacity—enhancing oxidative metabolism—and reduces ectopic fat accumulation in chronic obesity, a driver of impaired metabolic flexibility and muscle mass loss. Intramuscular adipose accumulation and defective mitochondrial function are features associated with obese-mediated muscle atrophy and hallmarks of neuromuscular disorders such as Duchenne muscular dystrophy. Therefore, miR-22 could represent a compelling molecular target to improve muscle health across various muscle-wasting conditions. This study describes a pharmacological strategy for the inhibition of miR-22 in skeletal muscle by employing a mixmer antisense oligonucleotide (ASO, anti-miR-22). Administration of the ASO in a mouse model of obesity positively modulated myogenesis while protecting dystrophic mice from muscle function decline, enhancing fatigue resistance, and limiting pathological fibrotic remodeling. Mechanistically, we show that anti-miR-22 treatment promotes derepression of genes involved in mitochondrial homeostasis, favoring oxidative fiber content regardless of the disease model, thus promoting a more resilient phenotype. Furthermore, we suggest that miR-22 inhibition increases autophagy by transcriptional activation of multiple negative regulators of mammalian target of rapamycin (mTOR) signaling to decrease immune infiltration and fibrosis. These findings position miR-22 as a promising therapeutic target for muscle atrophy and support its potential to restore muscle health.

## 1. Introduction

Skeletal muscle is a widely distributed tissue that comprises approximately 40% of the human body mass, with a significant contribution to different systemic processes [1]. In addition to generating mechanical force, skeletal muscles are responsible for the production and conversion of large amounts of chemical energy, deeply influencing body homeostasis, making skeletal muscle fundamental in controlling overall energy [2,3]. Muscle fibers are specialized to rely on a distinct bioenergetic program—glycolytic or oxidative—which determines their mechanical features, influencing the overall muscle functionality [4,5]. Elevated mitochondrial content and ATP production efficiency characterize oxidative fibers that are preferentially found in muscles employed in endurance tasks compared to glycolytic fibers that provide explosive force outputs [6]. Fiber-type plasticity is a hallmark of disease progression in several muscle-wasting conditions, and modulation of the fiber composition was suggested to help counteract muscle degeneration and preserve performance, particularly by promoting the maintenance or reprogramming toward oxidative fiber types [7,8]. Mitochondrial enrichment and a low ATP consumption rate provide slow-twitch type I oxidative fibers with better fatigue resistance, making them less susceptible to muscle atrophy and sarcopenia [9,10]. The increased proportion of this fiber type is observed in several myopathies, including Duchenne muscular dystrophy (DMD), which results from the selective degeneration of the fast-twitching fibers during disease progression [11,12].

DMD is an X-linked recessive genetic disorder caused by loss-of-function mutations in the dystrophin gene, compromising sarcolemmal integrity and leading to proximal muscle weakness and progressive terminal degeneration. Chronic muscle cell damage and the inflammatory milieu that results from the pathological phenotype promote extracellular matrix remodeling and replacement of muscle fibers with fibrotic and adipose tissues, further decreasing the contractile function of the muscle [13,14]. The intramuscular adipose tissue (IMAT) that results from this process is a depot primed for lipid storage that also promotes immune cell recruitment by the secretion of pro-inflammatory cytokines and chemokines, fostering the vicious cycle of dystrophies [15,16]. IMAT expansion is a general hallmark of muscle atrophy, characterizing age-related sarcopenia, cancer cachexia, and denervation, but also the pathogenesis of metabolic syndrome (MetS), a cluster of disorders with increasing incidence that elevate the risk of cardiovascular diseases and type 2 diabetes [17]. Despite DMD and MetS having different etiologies, DMD patients frequently exhibit cardiometabolic risk factors, such as obesity, dyslipidemia (including hypertriglyceridemia), insulin resistance, and glucose intolerance, mostly due to reduced motility and chronic glucocorticoid therapy [18].

Promotion of oxidative metabolism in skeletal muscles is a common tentative line of intervention for both conditions. In obesity, reduced reliance on oxidative metabolism for basal energy expenditure correlates with weight regain and development of insulin resistance [19,20]. Fatty acid oxidation (FAO) is also responsible for muscle remodeling, controlling the amount of slow-twitching oxidative fibers via regulation of the mitochondrial import of substrates for beta-oxidation through the carnitine shuttle system [21,22]. Furthermore, improving oxidative metabolism was confirmed as a possible therapeutic strategy for DMD by increasing mitochondrial biogenesis via peroxisome proliferator-activated receptor (PPAR) gamma coactivator 1-alpha (PGC-1α) activation. Enhanced expression and activation of PGC-1α reduce disease-related muscle injury and protect against fatigue in a dystrophic mouse model (mdx mice) and have also been shown to improve muscle strength in humans [23,24].

MicroRNAs (miRNAs) are a class of small non-coding RNAs involved in fine-tuning gene expression by base-pairing with target sites in protein-coding transcripts. The high evolutionary conservation of the miRNA target sites in the 3′ untranslated regions (3′-UTRs) of messenger RNAs and involvement in the progression of numerous diseases make them an increasingly important and versatile diagnostic and therapeutic tool [25,26,27,28]. In skeletal muscles, the regulation of genes involved in stem cell activation, myogenesis, fiber-type switch, and protein turnover is crucial to maintain muscle homeostasis in response to tissue injury and atrophy [29,30,31]. Therefore, the manipulation of miRNAs involved in the modulation of these signaling pathways has been investigated as a viable strategy to ameliorate the disease phenotype in sarcopenic obesity and muscular dystrophy. MicroRNA-22 (miR-22) is highly expressed in skeletal muscle and represents a metabolic miRNA involved in mitochondrial plasticity and lipogenesis that controls differentiation and fiber-type commitment in striated muscles. We previously showed that genetic ablation of miR-22 positively modulates energy expenditure, promoting enhanced oxidative capacity in the liver and adipose tissue of obese mice [32]. Furthermore, the same therapeutic effect was also confirmed through pharmacologic inhibition of miR-22, highlighting its translational potential as a drug target [33,34]. Therapeutic targeting of miRNAs is an expanding field, deploying chemically modified ASOs that feature improved target affinity and stability through a sequence of unmodified DNA nucleotides interspersed with high-affinity, nuclease-resistant locked nucleic acid (LNA) nucleotides.

In this study, we investigate the potential of miR-22 in skeletal muscle as a pharmacological target to rescue muscle wasting through positive rewiring of cellular metabolism in favor of oxidative metabolism and reduce lipid accumulation. We report that inhibition of miR-22 positively influences muscle endurance in dystrophic mice, promoting the enrichment of slow-oxidative type I fibers and increasing autophagosome formation through the inhibition of mTOR.

## 2. Results

### 2.1. Anti-miR-22 ASO Mediates miR-22 Inhibition in Muscles In Vitro and In Vivo

Given the metabolic benefits observed in liver and adipose tissue, we decided to investigate the effects of miR-22 inhibition in skeletal muscle as a therapeutic strategy to improve muscle wasting. The potency of an unconjugated ASO featuring LNA modifications with a mixmer design and a complete phosphorothioate (PS) backbone was initially tested in cell lines, both of murine and human origin. A chronic treatment of 96 h was performed on fully differentiated murine C2C12 and human primary skeletal muscle cells (HSkMC) using a 250 nM concentration of the ASO. Assessment of miR-22 levels via quantitative real-time (qRT) PCR upon treatment conclusion confirmed that the anti-miR-22 compound was highly potent, with comparable results across species, showing a silencing efficiency that exceeded 95% in both cell lines (log10FC in C2C12: −2.82 ± 0.03; log10FC in HSkMC: −1.75 ± 0.23) (Figure 1A). To further validate the pharmacologic efficacy, we checked the concurrent derepression of three metabolic transcripts (*Pten*, *Tet2*, *Sirt1*) harboring conserved miR-22 response elements in mouse and human [35,36,37], confirming consistent upregulation in both cell lines upon treatment (Figure 1B). As miR-22 was previously shown to exert post-transcriptional control of genes involved in oxidative cell metabolism, we assessed the expression of a panel of genes involved in mitochondrial maintenance (*Pparc1a*, *Nrf1*, *Opa1*), confirming their overall increased transcriptional activation (Figure 1C).

Next, we investigated the ASO in vivo to assess delivery to the skeletal muscle and select a therapeutic dose for functional studies. Subcutaneous administration of three doses (2.5/5/10 mg/Kg) of the ASO for 14 weeks was shown to mediate robust inhibition of miR-22 in the gastrocnemius (GAS) of C57BL/6J mice in a dose-dependent manner. Furthermore, the modulation of the direct targets previously tested in vitro was first shown to be inversely proportional to the level of free miR-22 (Figure 1D) in GAS, where a significant increase was proven at the highest dose. The highest dose at 10 mg/Kg was confirmed to also reduce the circulating levels of the targeted miRNA, proving the whole-body effect of the compound despite the local injection (Appendix A). Secondly, we directly quantified the accumulation of the ASO in GAS and obtained its relative quantification to the amount retained in the liver, where ASOs are primarily metabolized. The distribution profile of the compound showed a constant muscle-to-liver ratio regardless of dosage, averaging around 14-fold enrichment in the liver (Figure 1E), consistent with data from the subcutaneous administration of conjugated and unconjugated ASOs [38].

### 2.2. miR-22 Inhibition Promotes Transcriptional Activation of Myogenic Programs and Mitochondrial Homeostasis Regulation in Skeletal Muscles

Next, we evaluated whether miR-22 inhibition could improve muscle health in obesity. At the highest tested dose, the treatment produced a modulatory effect on body weight in a C57BL/6 mouse model of diet-induced obesity (DIO), promoting resistance to high-fat-diet-(HFD)-induced weight gain (Figure 2A), which is consistent with enhanced mitochondrial capacity. To elucidate the myocellular signaling pathways mainly involved in miR-22-mediated response to HFD, we performed a bulk RNA sequencing on GAS. Gene ontology analysis of the biological processes enriched during miR-22 silencing revealed downregulation of the post-synthetic macromolecular and post-translational protein modification (Appendix A), while profiling showed an enhanced myogenic signature associated with developmental and differentiation programs (Figure 2B). Evaluation of the expression profiles of individual differentially expressed genes (DEGs) confirmed a strong induction of myogenin (*Myog*) and myoblast determination protein 1 (*Myod1*) genes, involved in muscle growth and differentiation, which were further validated by qRT-PCR (Figure 2C). As RNAseq also uncovered a muscle contraction gene signature in treated mice (Appendix A), we performed a targeted gene expression analysis on structural genes to assess changes in the fiber type. Myosin heavy-chain isoform 7 (*Myh7*)—characteristic of slow-twitching oxidative fibers—emerged highly overexpressed in GAS upon treatment, without affecting the expression of the glycolytic-specific paralogue gene *Myh4* (Figure 2D). The observed fiber-type shift toward a more oxidative profile is supported by the upregulation of genes involved in mitochondrial health, which is consistent with what is shown in myotubes (Figure 2E).

### 2.3. miR-22 Inhibition Supports Oxidative Remodeling to Limit Fibrosis and Enhance Muscle Function in Muscular Dystrophy

The myogenic effect induced by miR-22 inhibition in vivo, alongside promoting a more oxidative fiber profile, holds therapeutic potential to counteract muscle wasting also in chronic myopathies. Since mitochondrial dysfunction and decreased oxidative metabolism are common features of muscle wasting in conditions such as DMD, we performed a bioinformatic analysis on publicly available miRNA-screening data to identify whether miR-22 is associated with muscle dystrophy. The analysis confirmed that circulating miR-22-3p is consistently elevated in four different murine models of muscle dystrophy alongside three muscle-specific miRNAs (myomiRs) involved in muscle regeneration (Appendix A). Therefore, we investigated whether chronic pharmacologic inhibition of miR-22 could ameliorate the dystrophic muscle phenotype of a murine model of DMD (D2-mdx) over 12 weeks (Figure 3A). An initial evaluation of the safety profile of the anti-mir-22 compound was performed via a histopathological examination of D2-mdx mice spleen, liver, and kidney, which confirmed the absence of splenomegaly (Figure 3B) as well as morphological alterations indicative of either hepatic (Figure 3C) or renal toxicity (Figure 3D), confirming the translational potential of the ASO in the dystrophic condition. Contrary to DIO mice, miR-22 inhibition did not influence the body weight in the D2-mdx model (Appendix A), nor did it exhibit muscle hypertrophy (Figure 3E). Successful inhibition of miR-22 was confirmed in both plasma (Appendix A) and striated muscles (Appendix A) of D2-mdx mice before evaluating the metabolic gene signature in dystrophic GAS. Targeted gene expression analysis confirmed the upregulation of several genes encoding transcriptional and post-transcriptional regulators of mitochondrial homeostasis in treated GAS, confirming that this gene signature, which improves the oxidative capacity, is mechanistically related to miR-22 silencing regardless of the disease model (Figure 3F). The enhanced oxidative transcriptional profile resulting from miR-22 inhibition was again observed in connection with a strong induction of type 1 fiber-specific *Myh7* gene (Figure 3G). However, despite favoring the oxidative fiber type, the treatment also positively affected fast glycolytic fibers in this model, possibly positively influencing the muscle contractile function. Therefore, we concurrently evaluated the functional readout of muscle performance throughout the study with longitudinal monitoring of muscle endurance and coordination, combining hanging and rotarod tests, alongside grip strength, which was separately measured (Figure 3A). All animals exhibited a general improvement in their motor functions during the first four weeks of the study, regardless of the treatment (Appendix A). A difference in muscle performance between the treatment groups was observed only starting from 14 weeks of age. Mice injected with the anti-miR-22 manifested progressively longer hanging times compared to the baseline and the control group, while the treatment was also shown to mitigate locomotor function decline induced by disease progression (Figure 3H). The impact of the anti-miR-22 therapy on the endurance capacity in dystrophic mice can be better captured by combining the performance times of the two tests that were sequentially performed, thus confirming that miR-22 inhibition promotes better resistance to overall muscle fatigue (Figure 3I) by stabilizing motor function in dystrophic muscle functional decline. A similar outcome trajectory to muscle endurance was also observed for grip strength, which the treatment showed to maintain during the disease course, in contrast to the drop observed in the control cohort (Figure 3J). Pharmacologic inhibition of miR-22 in the skeletal muscles of dystrophic mice, therefore, was shown to enhance muscle endurance and preserve contractile strength possibly through the support of mitochondrial homeostasis.

### 2.4. miR-22 Inhibition Limits Macrophage Infiltration in Dystrophic Skeletal Muscles by Inducing Autophagy Through mTOR Inhibition

To investigate the mechanism underlying the observed functional improvements, we assessed the impact of miR-22 in modulating the key hallmarks of muscular dystrophy. Intramuscular fat can negatively influence the contractile function of skeletal muscles. Oil red O (ORO) staining on GAS cryosections revealed a heterogeneous level of IMAT infiltration in this model, which resulted only in a non-statistically significant reduction in the median fat accumulation mediated by the treatment (Figure 4A). Reduced oxidative stress and adipose infiltration can also benefit the muscle by lowering tissue inflammation. Therefore, we performed F4/80 immunostaining to quantify macrophage infiltration in the fascicles in close proximity to the epimysium, since it is a primary site of immune cell infiltration by providing support for larger blood vessels. The treatment revealed a reduction in foci of macrophages, areas that were observed to be associated with increased cellular senescence (Figure 4B). Macrophages in these regions were also observed to express the beta galactosidase marker more frequently than in the rest of the muscle fibers, possibly contributing to impaired regeneration. To reconcile these observations, we queried the miRPathDB 2.0 database for experimentally confirmed target transcripts of miR-22 involved in senescence, identifying two negative regulators of the mechanistic target of rapamycin (mTOR) pathway, *Ddit4* and *Pten*. Counteraction of the pathological overactivation of mTOR in DMD is known to reduce tissue inflammation through the rescue of impaired autophagic flux. Therefore, we performed gene expression profiling on GAS and assessed overexpression of both targets in treated D2-mdx, as we previously identified in DIO mice (Figure 4C). Furthermore, to strengthen our hypothesis regarding improved autophagy induction, we investigated the expression of transcription factors involved in controlling several elements of the autophagic pathway. Indeed, forkhead box O family members, *FoxO1* and *FoxO3*, were both upregulated in treated animals (Appendix A). Increased autophagosome formation was then verified, assessing the levels of microtubule-associated protein 1 light chain 3 beta (MAP1LC3B) in the lipidated, membrane-bound form (LC3-II) over the cytosolic form (LC3-I). Improvement in autophagy is related to fibrosis. Picrosirius red staining (PSR) of GAS in dystrophic mice revealed a complementary reduction in connective tissue deposition following miR-22 inhibition only in three mice, highlighting subgroups of responders and non-responders (Figure 4E) among the treated animals. The separation according to treatment effect was supported by a negative correlation between fibrosis extension and the expression of genes associated with oxidative fibers, reconciling this key histological feature with metabolic rewiring. This suggests a beneficial impact of miR-22 inhibition on extracellular matrix remodeling during regeneration only in the presence of a higher oxidative metabolic profile (Figure 4F).

Collectively, our results demonstrate that the pharmacological inhibition of miR-22 is associated with enhanced autophagosome formation in dystrophic tissue and negative control of the mTOR signaling in skeletal muscle. These molecular alterations coincide with a marked reduction in immune cell infiltration and fibrosis, effects that are further supported by the induction of an oxidative fiber gene program.

## 3. Discussion

Muscle degeneration is a hallmark of numerous chronic conditions—including muscular dystrophies, sarcopenia, and metabolic diseases—and it is projected to rise globally with progressive aging of the population and increasing sedentary lifestyles. Modulation of several myomiRs, either supporting muscle regeneration or mitochondrial health, has already been proven successful to restore muscle function in models of muscle dystrophy [39,40,41]. Despite not being a myomiR, miR-22 is one of the most highly expressed miRNAs in striated muscles, where it has been previously described to have a role in myogenic differentiation, hypertrophy, and fiber-type remodeling [42,43,44]. Its marked tissue expression and poor cellular uptake and distribution of ASOs in the muscle tissue are challenging aspects that were initially addressed to prove whether effective pharmacologic targeting of miR-22 could be achieved. Although several strategies have been presented to improve ASO delivery to muscles, a selective delivery system for this tissue remains to be established, unlike for others [45], thus failing to avoid accumulation in the liver and kidneys while increasing molecular complexity and possibly reducing the stability of the compound. While primarily reaching the liver, the data presented in this paper show that the anti-miR-22 compound can readily engage with the target miRNA in skeletal muscle without requiring high doses while maintaining a weekly regimen. Inhibition of the miRNA upon subcutaneous administration was further observed in plasma, despite being less potent. However, the lower inhibition observed in circulating miR-22 could be explained by the large fraction of protein-bound miRNAs that are present in plasma, decreasing the binding efficiency of the ASO [46].

miR-22 is an emerging therapeutic target, as it is involved in the negative control of oxidative metabolism in liver and adipose tissue [33,47,48]. Among the main pathways regulated by miR-22, lipid and glucose metabolism, in conjunction with mitochondrial biogenesis, are collectively responsible for maintaining metabolic plasticity in muscle and were observed to be impaired in metabolic conditions such as obesity [48,49]. In a mouse model of obesity, we showed that miR-22 inhibition caused transcriptional activation of myogenic programs, with cohesive expression of muscle contraction and structure development genes, suggesting a beneficial adaptation of skeletal muscle as a compensatory mechanism in response to the HFD. Overexpression of genes such as *Myog*, *MyoD1*, and myosin heavy-chain isoforms reflects a transcriptional signature that increases force generation. Overexpression of *Myog* is known to favor an oxidative shift in muscle fibers, which better protects against muscle atrophy and improves resistance to fatigue [50,51]. Fiber-type switching upon miR-22 inhibition was observed in GAS, a muscle with a predominant glycolytic fiber composition, which underwent transcriptional upregulation of Myh7 compared to the HFD control, as well as for nuclear genes involved in the maintenance of mitochondrial homeostasis. Among them, Opa1 encodes for a crucial factor, orchestrating mitochondrial fusion, a paramount process in high-energy-demanding tissues to optimize maximal oxidative capacity [52,53]. Therefore, rescue of OPA1 in tandem with NRF1, a transcription factor whose expression and control over mitochondrial genome replication are dependent on PGC-1α, represents a cohesive effort to replenish the mitochondrial loss due to HFD.

Parallel to the induction of oxidative capacity and the support of mitochondrial fusion to improve muscle atrophy and wasting, mTOR signaling inhibition has been described as a promising therapeutic strategy for chronic degenerative conditions exhibiting mTOR complex 1 (mTORC1) hyperactivation, such as muscular dystrophy [54]. Despite the induction of catabolic processes associated with the expression of atrogenes and proteasomal degradation pathways, blockage of mTOR signaling benefits dystrophic muscles, decreasing fibrotic tissue and improving the autophagic flux, which normalizes protein turnover and lowers intracellular debris while decreasing inflammation [55,56]. miR-22 post-transcriptionally controls the expression of *Ddit4* and *Pten*, two genes encoding for negative regulators of mTORC1 that act on separate upstream pathways [57,58]. Therefore, D2-mdx mice were chosen to explore the therapeutic benefit of an anti-miR-22 therapy for the improved resemblance to the human disease pathophysiology, exhibiting a more severe fibrotic and inflammatory phenotype than the original dystrophic C57BL/10ScSn-Dmdmdx/J model. The disease course in the model is divided into three main phases, as previously characterized by Hammers et al. [59]. The treatment was started during the peak of the initial inflammatory phase, which is responsible for causing most of the muscle damage, thus maximizing the contribution of the following regenerative phase. As the beginning of the treatment was withheld until mice were 10 weeks old, this work shows the effects of miR-22 only during the secondary regenerative phase, which is characterized by the constant buildup of fibrotic tissue, which will be responsible for the final atrophy phase, set to start at 4 months of age. These aspects can be observed in control animals, whose locomotory performance peaked at 14 weeks old and declined afterward. Therefore, we used this timepoint to interpret the efficacy of the pharmacologic inhibition on motor function and muscle strength. The induction of mitochondrial remodeling in skeletal muscles toward increased oxidative metabolism, confirmed in both models, supports the beneficial effect on muscle endurance, as pharmacological and genetic induction of mitochondrial biogenesis has already been shown to produce increased locomotor function and fatigue resistance [60,61,62]. This further correlates with decreased accumulation of fibrotic tissue in GAS, which is concordant with the strong induction of oxidative fibers and mTORC1 inhibition [63,64]. Alongside fostering autophagy and mitochondrial biogenesis, prolonged inactivation of mTOR signaling is also responsible for the suppression of muscle hypertrophy and repair programs, leading to atrophy [62]. Neither of our models suffered muscle wasting nor muscle loss; we speculate that both the environmental and genetic stressors (prolonged HFD and defective DMD protein) cause hyperactivation of mTORC1 either by disrupting the feeding–fasting cycle or through chronic inflammation and consequent continuous activation of the regenerative programs. Dampening mTOR activation in this model would also reduce chronic inflammation, lowering immune cell infiltration. Supporting the immunogenic role of miR-22, miR-22 KO mice were shown to bear less tumor macrophage infiltration and reduce their activation, while miR-22 was separately confirmed to control monocyte/macrophage differentiation [65,66]. Therefore, systemic and not muscle-specific inhibition of miR-22 negatively impacts monocyte maturation, potentially decreasing pro-inflammatory cytokines and chemokines to reduce the chronic enhanced immune response induced by the continuous tissue damage.

The present study entails some methodological limitations. We acknowledge the absence of a chemistry-matched ASO with a scrambled sequence as a more appropriate matched control for the pharmacologic compound tested, although previous in vitro and in vivo observations suggested no significant difference between a non-targeting ASO and saline solution in modulating gene expression. Moreover, a more comprehensive investigation of the autophagic flux in these models is needed to confirm functional changes in the autophagic pathway following inhibition of miR-22.

Curative therapies for DMD—such as exon-skipping agents (e.g., Eteplirsen, Golodirsen, Viltolarsen, Casimersen, SRP-5051, PGN-ED051, ENTR-601-44) and gene therapies (e.g., Elevidys)—are advancing toward clinical use. We acknowledge that a miRNA-based approach like the one presented in this work could be best positioned as an adjuvant strategy. In this context, we propose that our anti-miR-22 ASO could be combined with exon-skipping therapies to enhance therapeutic efficacy. Future studies will focus on evaluating the feasibility and additive benefit of such a combinatorial approach. This work aimed to explore the broader therapeutic potential of miR-22 inhibition in mitigating multiple contributors to muscle wasting, leveraging its recent clinical progress in metabolic syndrome. Demonstrating a myogenic effect of miR-22 inhibition would support its application in weight-loss therapies to preserve lean mass, while potential weight-reducing properties may enhance its benefit in muscular dystrophy by improving metabolic status and quality of life.

This work suggests a regulatory role of miR-22 in the disease-dysregulated response causative of muscle atrophy in the context of obesity and muscular dystrophy. It proposes a novel therapeutic strategy based on ASO-mediated inhibition of the miRNA to improve muscle function through promoting a higher oxidative transcriptional profile and increased autophagy activation. Compensation for the disease-driven alterations was shown to decrease fibrotic accumulation and macrophage infiltration in fast glycolytic skeletal muscles, increasing muscle endurance and fatigue resistance. Functional and morphological improvement underscore the therapeutic value of targeting miR-22 to intervene upstream of broad degenerative cascades.

## 4. Materials and Methods

### 4.1. Characterization of the Antisense Nucleotide Compound

The anti-miR-22 ASO compound employed in this study was synthesized by Integrated DNA Technologies (IDT, Leuven, Belgium) with the sequence reported below:5′-CTTcaACtgGCAgCT-3′.(1)

The ASO was designed to feature a complete phosphorothioate backbone (i.e., all internucleoside linkages are PS) with interspersed DNA nucleotides (lowercase letters) and LNA-modified nucleotides (uppercase letters). The purity of the compound was ensured by HPLC purification and assessed via ESI-MS. The injectable solution was freshly prepared before each study, reconstructing the lyophilized compound in a 0.9% NaCl solution. The resulting 1 mg/mL stock solution was aliquoted and stored at −20 °C in single-use vials, thawed before the treatment. The concentration was calculated using the Lamber–Beer equation (MW: 5089.1 Da; ε = 134,000 L/(mole cm)). The estimated melting temperature of the compound is Tm = 66.1957 °C.

### 4.2. Cell Culture and Treatment

Murine C2C12 myoblasts (CRL-1772) and human primary skeletal muscle cells (HSkMC, PCS-950-010) were purchased from ATCC. Both cell lines were cultured at 37 °C, 5% CO_2_, following the manufacturer’s protocol. Briefly, C2C12 myoblasts were expanded in DMEM (Thermo Fisher, Cat. No. 11965092, Roskilde, Denmark) supplemented with 100 U/mL penicillin–streptomycin (Thermo Fisher, Cat. No. 15140122, Roskilde, Denmark) and 10% FBS (Sigma-Aldrich, Cat. No. 12103C, Darmstadt, Germany ) to 90% confluency. Cells were subsequently cultured in differentiation medium (DMEM, 2% FBS) for 5 days to induce the formation of syncytia. HSkMC was cultured in a Primary Skeletal Muscle Growth Kit (ATCC, Cat. No. PCS-950-040, Manassas, VA, USA). Cells were allowed to differentiate on fibronectin-coated plates in Skeletal Muscle Differentiation Medium (ATCC, Cat. No. PCS-950-050) for 5 days. Full differentiation was confirmed by morphology inspection for both cell lines. Differentiated myotubes were treated either with a vehicle solution (0.9% NaCl) or 250 nM anti-miR-22 ASO for 4 days. To maintain constant culture conditions, the medium was renewed every 2 days, freshly supplemented with the anti-miR-22 ASO.

### 4.3. Animals and Treatments

Handling of animals, used diets, housing, and animal experiments performed in this study were carried out in two separate models of muscle wasting in accordance with the “Guidelines for Animal Experimentation” and under approval of “The Danish Animal Experiments Inspectorate” (Protocol no 2019-15-0201-00310; 2024-15-0201-01702). In the first mouse model mediated by diet-induced obesity (DIO), 7-week-old C57BL/6JBomTac male mice (Taconic Biosciences, Leverkusen, Germany) were randomized in groups of 5 upon receipt and let acclimate for 2 weeks on a 12 h light–12 h dark cycle at 21–23 °C. Mice were subsequently switched to a 60% fat diet (HFD) (Brogaarden, Cat. No. D12492i, Lynge, Denmark) with ad libitum access to food and water and were weighed weekly (Appendix A). After 9 weeks, mice were randomized into control and treatment groups once they all surpassed 40 g, and they were subcutaneously injected with either sterile 0.9% saline solution (CTRL) or 10 mg/Kg/week of anti-miR-22 ASO for 14 weeks while on HFD. GAS muscles were collected from sacrificed animals, rinsed in cold PBS, and immediately frozen at −80 °C.

A second model of muscle atrophy featuring muscle dystrophy was obtained by purchasing 6-week-old D2.B10 (DBA/2-congenic) Dmd^mdx^ (D2-mdx) mice from Jackson Laboratory (Cat. No. 013141, Bar Harbor, ME, USA), later housed in a 12 h light–12 h dark cycle in a 21–23 °C pathogen-free barrier facility with free access to food (LabDiet 5K52, 6% fat) and water. Mice were left to acclimate for 4 weeks, randomized, and then subcutaneously injected with either 1% v:w of sterile 0.9% saline solution (CTRL) or 20 mg/Kg of anti-miR-22 ASO (loading dose). Injections were performed once weekly for a total of 12 weeks, during which mice were weighed twice a week (Appendix A). Starting from the second injection, the ASO was administered as half the loading dose for the remaining 11 weeks. All subcutaneous injections were performed with a 27 G needle to administer volumes of 10 mL/Kg, alternating right and left inguinal injections to improve animal welfare.

Mice from both studies were subjected to cardiac puncture under isoflurane sedation and subsequently sacrificed by cervical dislocation to be immediately dissected. Organs and tissues were harvested, weighed, and rinsed in cold PBS before being grossed and either snap frozen for RNA and protein extraction or fixed in 10% Neutral Buffered Formalin (NBF) for histological analysis.

### 4.4. Muscle Function Evaluation

To evaluate the effect of miR-22 inhibition on the dystrophic phenotype, tests were performed to assess balance, muscle endurance, and strength on D2-mdx mice, adapting protocols previously published [67]. Ten-week-old mice were initially trained for the rotarod test, forelimb strength test, and four-limb hanging test during the two days preceding the first injection. The training allowed the animals to familiarize themselves with the exercises and the experimenter. The tests were all performed every fourth week in the span of 2 days before the animals received their weekly dose. Mice were transferred to the test room and allowed to acclimate for 30 min. After being weighed, mice were challenged to hang with all four limbs from a metal grid (400 × 400 × 2 mm) as part of the lid of a rat cage. Singular mice were allowed to grasp the grid before being inverted 40 cm over a layer of soft fabric. The height was optimized to avoid intentional jumps while avoiding injuries. A maximum hanging time was set to 600 s, and mice falling prematurely were given up to three total attempts to improve their performance. This was performed to reduce the effect of clumsiness while avoiding overtiring the animal. Muscle endurance was calculated as impulse based on the formula:J = mg t,(2)
where mg is the weight of the mouse in newtons (N), and t is the longest hanging time in seconds. Impulse values were used for further analysis. All animals were allowed to rest for at least 1.5 h before being placed in a Rotamex-5 (Columbus Instruments, v2.20, Columbus, OH, USA) system. Mice in the same treatment group were challenged in groups of three to walk on the rolling spindle (3 cm diameter) with a starting rotation speed of 5 rpm, gradually accelerating (1.2 rpm/10 s) to 45 rpm. Every animal was allowed to repeat the run three times for up to 500 s, and the best running time was used for further analysis. Mice were returned to the housing facility and transferred back to the test room on the next day, 30 min prior to testing. After being weighed, mice were tested for grip strength by rapidly pulling the animal away from a bio-GS3 system (Bioseb, Pinellas Park, FL, USA), allowing them to register peak force once the front paws release the grid. Every animal was challenged with four sets of measurements, each set consisting of three rapid tests. Mice were allowed to rest for 90 s between sets, but not between each triplicate. The average of the four strongest pulls normalized on body weight was used for further analysis.

All the equipment was carefully sanitized with ethanol, and gloves were changed between animals to avoid any influence.

### 4.5. Blood Collection and Plasma Processing

Blood was obtained through cardiac puncture of anesthetized mice after 12 weeks of treatment. Blood was collected in K2EDTA microtainer tubes (BD 365975, Franklin Lakes, NJ, USA) and centrifuged at 2000× *g* for 15 min at 4 °C to collect the plasma for downstream analysis. The plasma was snap-frozen before storage at ultra-low temperature.

### 4.6. Histological Analysis

For histological analysis, kidneys, spleen, liver, GAS, and diaphragm muscles of D2-mdx mice were dissected and rinsed in cold PBS before fixing for 24 h at 4 °C. Muscle tissues were subsequently washed and incubated in 15% sucrose in PBS for 24 h before storing them in 30% sucrose for OCT freezing and cryo-sectioning. Kidneys, spleen, and liver were progressively dehydrated in 70% ethanol before paraffin embedding (FFPE). Samples were shipped to Histowiz, Inc. (Brooklyn, NY, USA) for embedding and staining. All organs and tissues were stained with hematoxylin and eosin (H&E) to assess tissue histology. Liver, kidney, and spleen were evaluated for retained structural physiology following ASO administration, focusing primarily on the absence of hyperplasic lesions, immune infiltration, and necrosis. Skeletal muscles were evaluated for fibrosis and lipid accumulation, immune infiltration, and cellular senescence. Muscular fibrosis was assessed with Picro Sirius Red (PSR) staining, whereas Oil Red-O (ORO) staining confirmed lipid accumulation on gastrocnemius 8 µm thick cryosections. To visualize the extent of immune infiltration and senescence, 5 µm thick sections of FFPE gastrocnemius muscles were deparaffinized in xylene and rehydrated before staining with anti-F4/80 (Cell Signaling Technology, #CST70076, Leiden, The Netherlands) and anti-beta-galactosidase (Ebioscience, #14-6773-81, Roskilde, Denmark) antibodies. The signal was visualized with 3,3′-diaminobenzidine (DAB) substrate, while nuclei were counterstained with hematoxylin. All acquisitions were performed on the Leica DM5000B brightfield microscope and were analyzed with Fiji software 1.54p (v. 1.8.0).

### 4.7. RNA Extraction, cDNA Preparation and Quantitative Real Time (qRT)-PCR

Total RNA was extracted from GAS and diaphragm muscles using the phenol–chloroform method as previously published [68]. Briefly, 20–30 mg of frozen muscle were manually homogenized with a mortar and pestle in 1 mL of QIAzol (Qiagen, Hilden, Germany) and incubated for 5 min at room temperature (RT). A total of 0.2 mL of chloroform (Sigma-Aldrich, Darmstadt, Germany) was added and thoroughly mixed with the lysate before centrifugation (12,000× *g*, 15 min, 4 °C). RNA was precipitated from the aqueous phase with 1:1 (*v*/*v*) cold isopropanol by centrifugation (12,000× *g*, 10 min, 4 °C) following 15 min RT incubation. The RNA pellet was washed twice with 0.5 mL of 70% ethanol and air-dried before being eluted in nuclease-free water. RNA was quantified with the Varioskan LUX Multimode Microplate Reader (Thermo Scientific, Roskilde, Denmark), and 2 µg of RNA was used for reverse transcription. miRNA quantification was performed, as previously described [69]. Briefly, total RNA was incubated with *E. coli* Poly(A) Polymerase (NEB# M0276) following the manufacturer’s protocol, and 10 µL of the reaction was reverse transcribed with SuperScript™ IV First-Strand Synthesis System (Invitrogen, Roskilde, Denmark), using a custom-made primer (5′-GGCCACGCGTCGACTAGTACTTTTTTTTTTTTTTTTTVN-3′) to enable elongation of the complementary DNA (cDNA). Diluted cDNA was quantified by qRT-PCR using PowerUp™ SYBR™ Green Master Mix (Applied Biosystems, A25742, Roskilde, Denmark) on a QuantStudio 6 Flex System (Applied Biosystems) following the manufacturer’s protocol. Gene expression was calculated with QuantStudio Real Time PCR software (v. 1.7.2) using the 2^ddCt^ method, using *Gapdh* and miR-16 as reference genes for transcripts and miR-22, respectively. Primers were purchased from Integrated DNA Technology; sequences are provided in Table 1.

### 4.8. ELISA for Quantification of Anti-miR-22 ASO in Liver and Skeletal Muscle

Snap-frozen liver and GAS samples (~50 mg) of control or anti-miR-22 ASO-treated DIO mice were homogenized in 700 µL of RLT buffer (Qiagen). Homogenates were diluted 1:2 in RLT and centrifuged (17,000× *g*, 20 min, 4 °C) to collect the supernatant for a downstream analysis.

The ASO level in the tissues was quantified using a custom hybridization ELISA (hELISA) with biotinylated DNA/LNA probes. A probe pair was optimized to hybridize two distinct parts of the ASO sequence while featuring either a biotin or digoxin molecule positioned at the 5′ or 3′ end, respectively.

Diluted samples and standard curves were prepared in triplicate by mixing 50 µL of the sample or standard with 50 µL of a 25 nM probe mix in a 10× SCCT buffer containing 0.1% Tween-20. After 1 h of incubation at room temperature on gentle agitation (300 rpm), the hybridized complexes were transferred to streptavidin-coated plates for 30 min to immobilize the hybrid. Plates were washed and incubated with anti-digoxigenin–peroxidase Fab fragments (Roche, 1:3000, Basel, Switzerland). Detection was performed by incubating the immobilized samples for 4 min with a TMB substrate. The reaction was stopped with 0.5 M sulfuric acid, and absorbance was measured at 450 nm.

### 4.9. Library Preparation and Sequencing

To unravel the primary effect of miR-22 inhibition on the transcriptome of skeletal muscles, we performed a bulk RNA-seq analysis on RNA extracted from the gastrocnemius of DIO mice fed 60% HFD. RNA quality was checked using the TapeStation (Agilent, Waldbronn, Germany), allowing for a minimum RIN value of 7. 500 ng of RNA to be used for library preparation using the Illumina stranded mRNA prep kit (Illumina, Cambridge, UK), following the manufacturer’s instructions.

Libraries were assessed for quality using the TapeStation (Agilent), showing a peak at around 280 bp for all samples, indicating good library quality. The libraries were then quantified using the Qubit 1X dsDNA High Sensitivity kit (Invitrogen) and sequenced on the Illumina NextSeq1000 System (paired-end 60 bp reads). RNA-seq and library preparation were performed at the Molecular Biotechnology Center (MBC) of Torino University.

### 4.10. RNA Sequencing Analysis

Sequencing data were pre-processed using a custom Snakemake pipeline. Briefly, the pipeline executes a set of rules in order to conduct the following:I.Preprocess and perform quality control for sequencing data using the FastQC tool (v. 0.12.0) (https://www.bioinformatics.babraham.ac.uk/projects/fastqc/, accessed on 8 October 2025) and the MultiQC framework (v. 1.29) (https://github.com/MultiQC/MultiQC, accessed on 8 October 2025).II.Trimming of the sequencing reads for adapter/low-quality sequence using the fastp tool (v. 1.0.0) (https://github.com/OpenGene/fastp, accessed on 8 October 2025) with default parameters.III.Mapping the filtered reads to GRcm39 genome using STAR with the following options: --outSAMtype BAM SortedByCoordinate --outSAMunmapped Within --chimOutType WithinBAM --outFilterType BySJout --outFilterMultimapNmax 200 --alignSJoverhangMin 8 --alignSJDBoverhangMin 1 --outFilterMismatchNmax 999 --outFilterMismatchNoverReadLmax 0.04 --alignIntronMin 20 --alignIntronMax 1,000,000 --alignMatesGapMaxIV.Quantify the number of reads mapping to each gene using gene annotation from Gencode M25 using the FeatureCounts utility of the Subread package [70] with parameters set to -t exon --extraAttributes gene_name -p -C --countReadPairs. Counts were restricted to the last 500 bp of the 3′UTR of each gene to handle biases in sequencing coverage.V.Perform read counts normalization and differential gene expression analysis using DESeq2 [71] with its default settings to fit the negative binomial model and perform Wald tests, identifying genes that are differentially expressed between the anti-miR-22 ASO-treated and control muscles. After calculating the initial results, the lfcShrink function of DESEq2 with the “ashr” method was used to moderate extreme Log Fold Change estimates, reducing the impact of low-count genes and leading to more reliable effect size estimates for ranking and interpretation.

### 4.11. Protein Extraction and Immunoblotting

Snap-frozen muscles were homogenized with Tissuelyser III (Qiagen) in ice-cold lysis buffer (20 mM Tris-HCl, pH 8.0, 137 mM NaCl, 2.7 mM KCl, 1 mM MgCl_2_, 1 mM NaEDTA, 1 mM EGTA, 1% Triton X-100, 10% glycerol) supplemented with fresh phosphatase and protease inhibitors (Roche, # 04693116001, Basel, Switzerland). Lysates were centrifuged at 13,000× *g* for 20 min at 4 °C and quantified with Protein Assay Dye Reagent Concentrate (Bio-Rad, #5000006, Basel, Switzerland) following the manufacturer’s protocol. In total, 20 µg of protein lysates were separated by SDS-PAGE on 4–12% gradient Bis-Tris gels (ThermoFisher Scientific, #NW04120BOX, Roskilde, Denmark) and transferred to a nitrocellulose membrane on a Trans-Blot Turbo system (Bio-Rad, Basel, Switzerland). The membrane was blocked in 5% non-fat dry milk (Santa Cruz Biotechnology, Heidelberg, Germany) in 0.1% TBS-Tween (TBS-T) for 1 h before overnight incubation with primary antibodies in 3% BSA TBS-T (1:5000 anti-GAPDH (Protein Tech, #60004-1-Ig, Manchester, UK), 1:200 anti-LC3B (Santa Cruz, #SC-376404)). The membranes were washed three times for 10 min with wash buffer (TBS-T) before and after incubation with the appropriate secondary antibody (Cell Signaling Technology, #7076). The signal was imaged on a ChemiDoc MP instrument (Bio-Rad) upon incubation with Clarity Western ECL substrate (Bio-Rad). The band signal was quantified with Image Lab v.6.1 software (Bio-Rad). The LC3 band signal was normalized on GAPDH first, and the LC3 II/I ratio was calculated on the normalized band signals.

### 4.12. Statistical Analysis

All data represent at least three independent experiments unless specified. Post hoc statistical analysis was performed using Graphpad Prism v.10.4 and encompassed unpaired *t*-test corrected with the Holm-Šidak method for multiple comparisons, and mixed-effects models for repeated measurements with the alpha threshold set at 0.05 and corrected for multiple comparisons based on the Tukey test. One outlier in Figure 1D was removed, as it fell outside the 97.5% tolerance interval of the bivariate dataset obtained by plotting the first two PCA components of all biological replicates within the 10 mg/Kg treatment group. PCA components for each biological replicate were calculated with the *PCAGrid* function (“rrcov” R package v. 1.7-7) using the gene expression data of nine genes and graphed on a biplot to confirm via visual inspection.

## Figures and Tables

**Figure 1 ijms-26-09900-f001:**
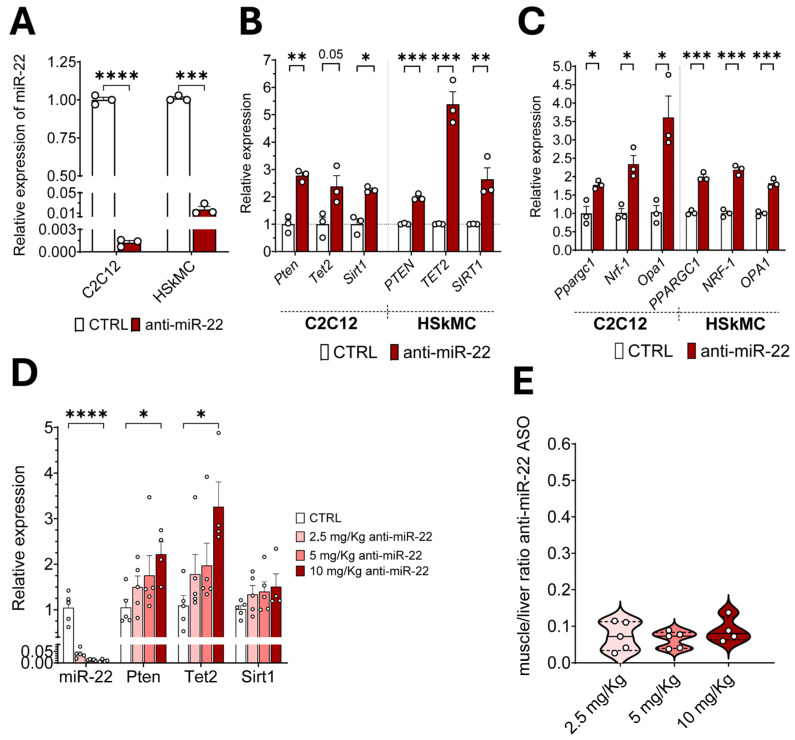
Anti-miR-22 mediates miR-22 inhibition in vitro and in vivo. qRT-PCR assessment of miR-22 levels (**A**) and expression of direct target genes (**B**) in murine C2C12 myotubes (*n* = 3) and differentiated human skeletal muscle cells (HSkMC) (*n* = 3) after a 96 h treatment with either saline solution (CTRL) or 250 nM anti-miR-22 ASO. (**C**) Gene expression analysis of genes involved in mitochondrial biogenesis (*Pparg1*, *Nrf1*) and mitochondrial fusion (*Opa1*) in C2C12 cells (*n* = 3) and HSKMC (*n* = 3) cells. (**D**) Level of miR-22 and expression of its direct target genes at three concentrations of anti-miR-22 ASO (shades of red). RNA was extracted from the GAS of an HFD-induced obesity mouse model (60% fat) (*n* = 5). (**E**) Direct quantification of the anti-miR-22 ASO in muscle relative to liver accumulation at the three working doses. Median (Q2) (solid line) with first (Q1) and third quartile (Q3) (dotted lines) (*n* = 5). Data are otherwise shown as mean ± SEM. * *p* < 0.05, ** *p* < 0.01, *** *p* < 0.001, **** *p* < 0.0001 (unpaired *t*-test).

**Figure 2 ijms-26-09900-f002:**
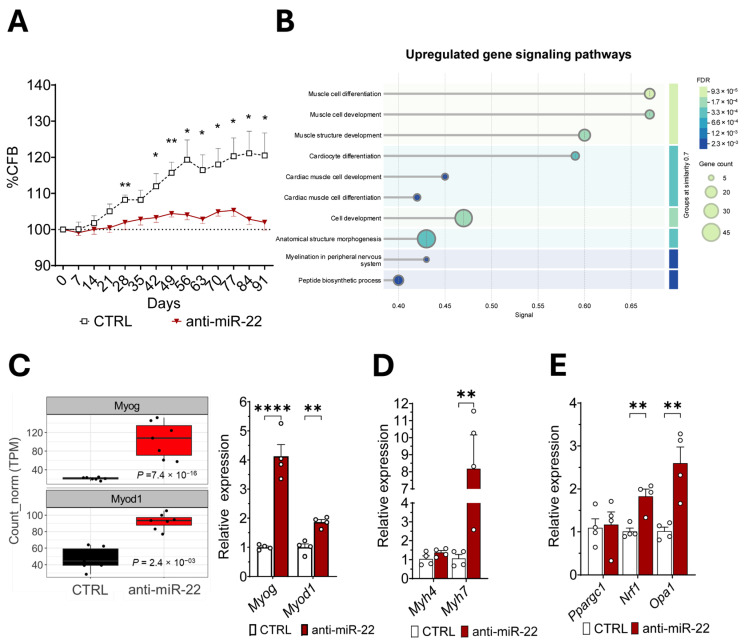
miR-22 inhibition drives muscular remodeling through mitochondrial biogenesis and fusion in healthy muscles. (**A**) Percentage of change from baseline (%CFB) in the weight of DIO mice injected with either control or anti-miR-22 ASO at 10 mg/Kg (*n* = 5). (**B**) Gene ontology analysis of upregulated biological processes enriched for differentially expressed genes (DEG) induced by anti-miR-22 ASO in GAS muscle of DIO mice. (**C**) Transcript counts of key genes differentially expressed in the treated animals compared to the control and involved in myogenesis and muscle differentiation (TPM); their validation by qRT-PCR (right). (**D**) Comparison between expression of slow (*Myh7*) and fast (*Myh4*) myosin heavy-chain genes in GAS of DIO mice. (**E**) Gene expression analysis of genes involved in mitochondrial biogenesis (*Pparg1*, *Nrf1*) and mitochondrial fusion (*Opa1*) in DIO mice GAS (*n* = 5). Data are shown as mean ± SEM. * *p* < 0.05, ** *p* < 0.01, **** *p* < 0.0001 (mixed-effects models (**A**); unpaired *t*-test (**C**–**E**)).

**Figure 3 ijms-26-09900-f003:**
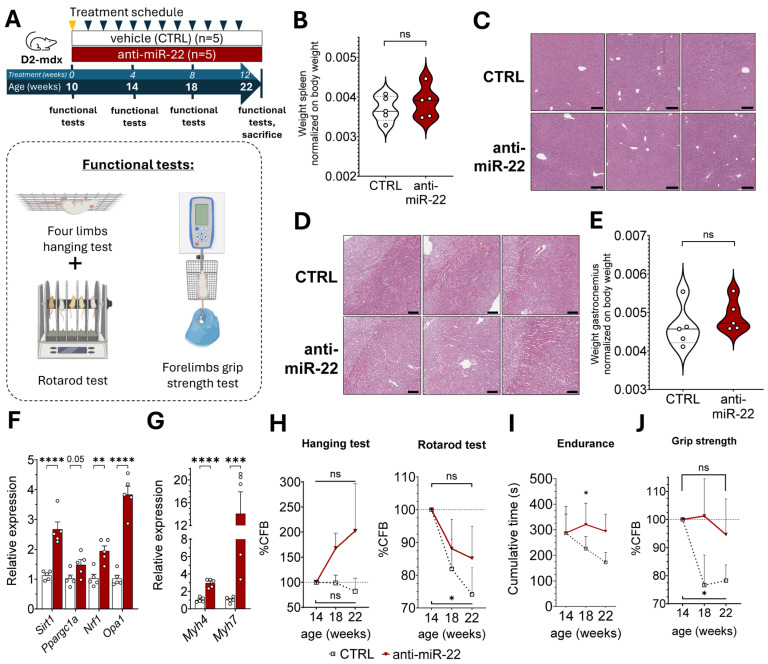
miR-22 inhibition improves the dystrophic phenotype in a DMD mouse model, ameliorating motor functions and fibrosis in limb muscles. (**A**) Schematic representation of the study design with the motor function tests performed during the study. D2-mdx mice were injected weekly (arrows) either with saline solution (CTRL) or the anti-miR-22 ASO at 10 mg/Kg for 12 weeks. Loading dose (yellow arrow) was 20 mg/Kg. Motor function was assessed every four weeks from the beginning of the study. (**B**) Spleen-to-body ratio in control and ASO-treated mice (Q1, Q2, Q3) (*n* = 5). Representative H&E staining of livers (**C**) and kidneys (**D**) of control and ASO-treated D2-mdx mice employed in histological analysis for assessment of the ASO’s possible hepatotoxicity and nephrotoxicity (*n* = 5) (bar = 200 µm). (**E**) Muscle-to-body ratio of GAS in control over ASO-treated mice (Q1, Q2, Q3). (**F**) Expression of genes associated with mitochondrial biogenesis and fusion in the GAS muscle of treated and control D2-mdx mice. (**G**) Comparison between expression of slow (*Myh7*) and fast (*Myh4*) myosin heavy chain genes in GAS of control and ASO-treated D2-mdx mice. The effect of miR-22 inhibition on muscle endurance and balance was assessed as the percentage of change from baseline (%CFB) in the last 8 weeks of the study in the four-limbs-hanging test and rotarod test (**H**), while muscle strength (**J**) was quantified by the forelimb grip strength test. (**I**) Cumulative time of hanging and walking time measured in the hanging and rotarod tests performed subsequently are plotted as a measure of muscle endurance. Data are shown as mean ± SEM unless specified. ns: *p* > 0.05, * *p* < 0.05, ** *p* < 0.01, *** *p* < 0.001, **** *p* < 0.0001 (unpaired *t*-test (**B**,**E**–**G**); mixed-effects models (**H**–**J**)).

**Figure 4 ijms-26-09900-f004:**
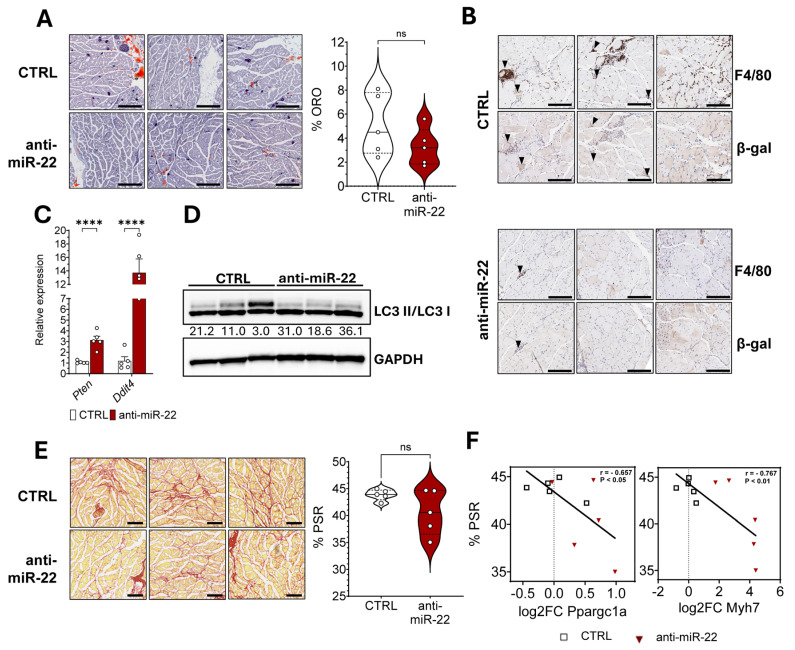
Molecular and functional improvement mediated by miR-22 inhibition is promoted through decreased immune infiltration due to enhanced autophagy in skeletal muscles of D2-mdx mice. (**A**) Oil red O (ORO) staining and quantification of intramuscular lipid accumulation in GAS of control and treated mice (Q1, Q2, Q3) (*n* = 5) (bar = 500 µm). (**B**) Immunohistochemical staining (DAB) performed on histologically matched regions on serial consecutive tissue sections to visualize macrophage infiltration (F4/80) and cellular senescence (β-gal) in GAS of control (upper panel) and anti-miR-22 ASO-treated mice (lower panel) (*n* = 3) (bar = 200 µm). Black arrows highlight senescent macrophages in foci. (**C**) Gene expression of repressors of the mTOR signaling pathway in GAS of control and ASO-treated mice (*n* = 5). (**D**) Immunoblot quantification of autophagic activation in GAS is calculated as the ratio of lipidated MAP1LC3B II and cytosolic MAP1LC3B I forms, normalized over GAPDH expression. Ratios are provided below the image (*n* = 3). (**E**) Picrosirius red staining (PSR) and quantification of connective and fibrotic tissue in GAS muscles of control and anti-miR-22 ASO-treated mice (Q1, Q2, Q3) (*n* = 5) (bar = 200 µm). (**F**) Correlation between *Pparg1a* gene expression and fibrosis in GAS of D2-mdx mice. Data are shown as mean ± SEM unless specified. ns: *p* > 0.05, **** *p* < 0.0001 (unpaired *t*-test (**A**,**C**,**E**); Pearson correlation, two-tailed (**F**)).

**Table 1 ijms-26-09900-t001:** Primer sequences used for gene expression analysis.

Gene	Species	Forward Primer	Reverse Primer
*Ddit4*	mouse	GCCGGAGGAAGACTCCTCATA	CATCAGGTTGGCACACAGGT
*Fbxo32*	mouse	CGACCTGCCTGTGTGCTTAC	CTTGCGAATCTGCCTCTCTG
*Foxo1*	mouse	CGTGCCCTACTTCAAGGATAA	GCACTCGAATAAACTTGCTGTG
*Foxo3*	mouse	GTGTGCCCTACTTCAAGGATAA	TCATTCTGAACGCGCATGA
*Gapdh*	mouse	TGACCACAGTCCATGCCATC	GACGGACACATTGGGGGTAG
*Myh4*	mouse	GCCTCCTTCTTCATCTGGTAA	CGATTCGCTCCTTTTCAGAC
*Myh7*	mouse	TACTTGCTACCCTCAGGTGG	ATGGCTGAGCCTTGGATTCTC
*Myod1*	mouse	TACGACACCGCCTACTACA	GGAGATGCGCTCCACTATG
*Myog*	mouse	AGTGAATGCAACTCCCACAG	GACGTAAGGGAGTGCAGATTG
*Nrf1*	mouse	ACAGATAGTCCTGTCTGGGGAAA	TGGTACATGCTCACAGGGATCT
*Opa1*	mouse	CCGACCTGGACAAGATTACTG	CCATGATCTGTTGCTCGAAATG
*Ppargc1*	mouse	CCCATACACAACCGCAGTC	GAACCCTTGGGGTCATTTG
*Pten*	mouse	AGGCACAAGAGGCCCTAGAT	CTGACTGGGAATTGTGACTCC
*Sirt1*	mouse	CAGTGAGAAAATGCTGGCCTA	TTGGTGGTACAAACAGGTATTGA
*Tet2*	mouse	GTGGACTGCGAGGCTGAG	AGTCTTGGGAGGGCAAGC
*Trim63*	mouse	GGTGCCTACTTGCTCCTTGT	CTGGTGGCTATTCTCCTTGG
*GAPDH*	human	GATTCCACCCATGGCAAATTC	GTCATGAGTCCTTCCACGATAC
*NRF1*	human	GTATCTCACCCTCCAAACCTAAC	CCAGGATCATGCTCTTGTACTT
*OPA1*	human	CTCACCATGTGGCCCTATTT	ACGGTACAGCCTTCTTTCAC
*PPARGC1*	human	ACGAAGAGCTCTCCTCCTTC	CAGCATAGAGTTGCTCCTCC
*PTEN*	human	TCCACAAACAGAACAAGATGCTA	CGATTTCTTGATCACATAGACTTCC
*SIRT1*	human	AAATGCTGGCCTAATAGAGTGG	TGGCAAAAACAGATACTGATTACC
*TET2*	human	GAAAAAGATGAAGGTCCTTTTTATACC	TTTACCCTTCTGTCCAAACCTT

## Data Availability

The datasets generated and/or analyzed during the current study are available from the corresponding authors on reasonable request. The RNA-seq dataset discussed in this publication has been deposited in NCBI’s Gene Expression Omnibus [72] and is accessible through GEO Series accession number GSE309212 (https://www.ncbi.nlm.nih.gov/geo/query/acc.cgi?acc=GSE309212, accessed on 8 October 2025).

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
