# Peer review of "microRNA-22 Inhibition Stimulates Mitochondrial Homeostasis and Intracellular Degradation Pathways to Prevent Muscle Wasting"

_ijms, 2025, doi:10.3390/ijms26209900_

Round 1
Reviewer 1 Report
Comments and Suggestions for Authors
The article by Tomasini describes inhibition of miR-22 as a potential therapeutic approach for treating myopathies. While the work performed is backed up by solid evidence of data, there are some lingering questions that contradict the current literature. Please see my comments below:
- What is the need for high fat diet in this study which focuses on myopathies and not obesity? A high fat diet increases miR-22 levels, especially in liver and adipose tissues. It also has been shown that a loss of miR-22 in skeletal muscles in high fat induced diet protects them from the negative effect, which is exactly this manuscript does through miR-22 silencing. But the need for HFD in the first place is not fully understandable. Please comment.
- As the authors have mentioned in the manuscript, miR-22 is one of the many miRNAs along with myomiRs that is upregulated in myopathies, and thus in D2-mdx mice. However, miR-22-3p has been well established to promote myoblast differentiation and muscle repair through HDAC4 inhibition – which is exactly what happens in a myopathic state in these mdx mice. While miR-22-3p is expressed more highly in fast-twitch (glycolytic) muscles, its function in promoting differentiation and regeneration is not limited to that fiber type. Please comment on it and the apparent contradiction of trying to lower miR-22 levels to reverse muscle pathology. Please weave in the role of HFD in this conversation as well.
- If possible, please provide more information on the anti-miR ASO – the length, distribution of LNA residues within the sequence, melt-temp analysis.
- Please add the details like ASO concentration and the day RT-qPCR was done for Figure 1A in text and figure caption. The method section says the cells were treated for 4 days with the ASO but it also mentions the media was replaced every 2 days. Does it mean, the cells were supplemented with fresh ASO during the media change? Please clarify.
- For discussion of the results of Figure 1D/E, please mention within text that ASOs were administered for 14 weeks.
- For Figure 1D: Are some of the outliers simply shown and not considered for plotting the bar graphs?
- Why was GAS chosen as the skeletal muscle of choice over TA or Quadriceps?
- Figure 3: Please upload high resolution images of the kidney and liver histopath. It is difficult to view differences these small low-resolution images.
Author Response
Summary
We would like to sincerely thank the reviewer for the objective and thorough evaluation of our manuscript. Below, we provide a detailed response to each of the points raised. We believe that we have adequately addressed all the comments and recommendations received, mainly focused on reconciling our work with the current literature. Therefore, we hope that the revised manuscript will now be considered suitable for publication in the International Journal of Molecular Sciences.
Comment 1: What is the need for high fat diet in this study which focuses on myopathies and not obesity? A high fat diet increases miR-22 levels, especially in liver and adipose tissues. It also has been shown that a loss of miR-22 in skeletal muscles in high fat induced diet protects them from the negative effect, which is exactly this manuscript does through miR-22 silencing. But the need for HFD in the first place is not fully understandable. Please comment.
Response 1: Thank you for your question. In this study we aim at investigating the role of miR-22 as therapeutic target for muscle wasting. Therefore, we have studied the effect of pharmacologic inhibition of miR-22 in two models of muscle wasting, one induced by environmental exposure (DIO), the second induced by genetic cause (D2-mdx). Obesity is responsible for muscle dysfunction, favoring the expansion of glycolytic fibers at the expense of low oxidative fibers, and impairing mitochondrial efficiency in energy production. Cross talk between skeletal muscles and dysfunctional adipocytes present in the enlarged visceral adipose tissue induced in obesity was confirmed to mediate muscle atrophy negatively modulating the expression of several myogenic genes such as MyoD1 and MYOG (10.2337/db14-0796). Furthermore, HFD is used to generate models of muscle atrophy in rodents (10.1016/j.archger.2023.105025, 10.3390/antiox10101633, 10.1007/s40520-015-0316-5).
Comment 2: As the authors have mentioned in the manuscript, miR-22 is one of the many miRNAs along with myomiRs that is upregulated in myopathies, and thus in D2-mdx mice. However, miR-22-3p has been well established to promote myoblast differentiation and muscle repair through HDAC4 inhibition – which is exactly what happens in a myopathic state in these mdx mice. While miR-22-3p is expressed more highly in fast-twitch (glycolytic) muscles, its function in promoting differentiation and regeneration is not limited to that fiber type. Please comment on it and the apparent contradiction of trying to lower miR-22 levels to reverse muscle pathology. Please weave in the role of HFD in this conversation as well.
Response 2: Thank you for raising this concern. We understand that the role of miR-22 is controversial in maintaining muscle homeostasis. Although pan HDAC inhibitors such as Givinostat have been successfully proven to improve muscle function acting on fibrosis and regeneration in DMD, our hypothesis is based on decreasing the inflammatory response that is responsible for muscle wasting by improving macroautophagy and mitochondrial health. Limiting pathological hyperactivation of signaling pathways related to muscle growth such as mTOR signaling positively was shown to positively affect muscle strength improving the autophagic flux in muscles. whereas its complete inhibition induces the expression of atrogenes and the development of muscle cachexia. Although our work did not focus on IR, miR-22 inhibition has been shown to improve metabolic flexibility and reduce IR, a risk factor that leads to the secretion of pro-inflammatory cytokines and chemokines that worsen muscle wasting in the obese phenotype. Finally, miR-22 inhibition has been already characterized to lower hepatic lipid accumulation and inflammation in MAFLD, pathophysiological aspects shared by DMD. Therefore, we wanted to investigate and test if miR-22 inhibition could have similar effect in muscle, another key tissue for glucose metabolism.
Comment 3: If possible, please provide more information on the anti-miR ASO – the length, distribution of LNA residues within the sequence, melt-temp analysis.
Response 3:. We have now moved the sequence and physicochemical properties of the ASO – previously stated within the “Cell culture and treatment” section – in the stand-alone section 4.1 (Lines 435-447) to avoid confusion and improve clarity.
Comment 4: Please add the details like ASO concentration and the day RT-qPCR was done for Figure 1A in text and figure caption. The method section says the cells were treated for 4 days with the ASO but it also mentions the media was replaced every 2 days. Does it mean, the cells were supplemented with fresh ASO during the media change? Please clarify.
Response 4: We thank you for your comment. Caption and text section regarding Figure 1A contain now the information 250 µM anti-miR-22 ASO to improve immediate understanding. Material and methods section that refer to cell treatment was implemented with the phrase: “To maintain constant culture conditions, the medium was renewed every 2 days, freshly supplemented with the anti-miR-22 ASO” (lines 461-462).
Comment 5: For discussion of the results of Figure 1D/E, please mention within text that ASOs were administered for 14 weeks.
Response 5: Thank you for the suggestion. We now added “for 14 weeks” in Line 140.
Comment 6: For Figure 1D: Are some of the outliers simply shown and not considered for plotting the bar graphs?
Response 6: Thank you for raising this concern. All data shown are included in computing the averages represented in the bar plots. Outliers were removed before plotting the data when falling outside the 97,5% tolerance interval of the bivariate data set obtained plotting the first two PCA components of all biological replicates within each treatment group. PCA components for each biological replicate were calculated by PCAGrid function (“rrcov” R package) using gene expression data of 9 genes and graphed on a biplot also to confirm by visual inspection. We have inserted this information in section 4.12 of the material and methods.
Comment 7: Why was GAS chosen as the skeletal muscle of choice over TA or Quadriceps?
Response 7: We decided to focus on the gastrocnemius muscles as they feature higher glycolytic fiber composition. Glycolytic fiber switch is a recognized pathological feature of obesity and leads to faster development of muscular fatigue. Analysis of this muscle allows to better characterize the effect of enhancing oxidative fibers by boosting oxidative programs and mitochondrial function through anti-miR-22 therapy. Increased synthesis of PGC-1α, OPA1 and OXPHOS complexes has been previously observed in the gastrocnemius muscles of D2-mdx mice following aerobic exercise, which is known to improve oxidative metabolism in skeletal muscles (10.1113/JP286768).
Comment 8: Figure 3: Please upload high resolution images of the kidney and liver histopath. It is difficult to view differences these small low-resolution images.
Response 8: We apologize for the low resolution of the images. We have inserted in the main text new images and formats that would allow a higher resolution. The manuscript is provided both as word and PDF files to avoid similar issues.
Reviewer 2 Report
Comments and Suggestions for Authors
This manuscript investigates whether pharmacologic inhibition of miR‑22 with an LNA‑mixmer antisense oligonucleotide (ASO) improves skeletal muscle health across two contexts: (i) diet‑induced obesity (DIO) and (ii) dystrophic D2‑mdx mice. The authors reported robust miR‑22 knockdown in vitro and in vivo, up‑regulation of genes associated with mitochondrial biogenesis/fusion and oxidative fibre programs (Myh7, PGC‑1α/NRF1/OPA1), improved fatigue resistance and grip strength in D2‑mdx, and reduced macrophage infiltration and fibrosis, which they link mechanistically to mTOR inhibition and enhanced autophagy. The work aligns with the journal’s scope and will interest readers in muscle biology, miRNA therapeutics, and neuromuscular disease.
-
Major comments:
- In vitro and in vivo studies use saline controls only. A sequence‑mismatched, chemistry‑matched LNA/DNA PS control (scrambled or seed‑mismatch) is needed to distinguish class effects from on‑target activity. Please include (or acknowledge as a limitation) and report its sequence, purity, and performance.
- RNA‑seq reporting and data deposition: missing n/group, mean read depth, mapping %/QC metrics, DESeq2 thresholds (BH‑FDR), and GO analysis tool/parameters.
- LC3‑II/I ratio alone does not establish flux. Please include flux assays (e.g., bafilomycin/chloroquine pretreatment in vivo where feasible or ex vivo), and quantify p62/SQSTM1, with timing/dose reported. Alternatively, clearly acknowledge this limitation and temper causal statements.
- ASO characterization and dosing should include information on vendor, purity, injection volume (mL/kg), site, and needle gauge.
- The dataset and relevant RNAseq data should be deposited into an online repository.
- Fig 1A n =2 is in sufficient power for statistical analysis.
- Figure fonts are too small to be read.
Specific comment:
- Line 23, “ectopic fat ACCUMULATION in chronic obesity”
- Line 38-39, “These findings position miR-22 as a promising therapeutic target for muscle-wasting conditions”, this is questionable as not all conditions are the same, such as sarcopenia and cachexia are both different conditions to that of DMD.
- Line 47, “mechanical force, THESE muscles…”
- Line 58-62, suggest to rewrite to improve clarity, consider “Enriched mitochondrial content and a low ATP consumption rate provide slow-twitch type I oxidative fibers with superior fatigue resistance, making them less susceptible to muscle atrophy and sarcopenia. An increased proportion of this fibre type is observed in several myopathies, including Duchenne muscular dystrophy (DMD), as a result of the selective degeneration of fast-twitch fibres during disease progression.”
- Line 76-78, needs clarification as DMD is primarily a neuromuscular disease, while MetS is a lifestyle related, consider “Although DMD and MetS have different aetiologies, DMD patients frequently exhibit cardiometabolic complications such as obesity, dyslipidemia (including hypertriglyceridemia), and glucose intolerance or insulin resistance, largely due to reduced mobility and chronic glucocorticoid therapy.”
- Line 82, FAO is not defined.
- Line 89, “while increasing muscle strength also in humans) change to “and have also been shown to improve muscle strength in humans”.
- Line 96, should be “In SKELETAL muscles”.
- Line 115, would “preservation” be more accurate than “enrichment”?
- Line 142, change “tissue” to “GAS”.
- Line 143, change “10 mg/Kg dose” to “dose at 10 mg/Kg”.
- Line 147-149, if the muscle-to-liver ratio of the ASO is not changed by the different dosage, what might be responsible for the dose-dependent effect seen in Figure 1 D?
- Line 171, “while profiling SHOWED an enhanced myogenic signature.
- Fig 2A, what is the reason for presenting the %CFB rather than the actual body weight data (and throughout the manuscript)?
- Fig 2C, please add y-axis title and unit
- Line 267, minor beneficial effect was claimed about miR-22 inhibition on IMAT, while no statistical significance was indicated on Fig 4A.
- Line 291-292, a complementary reduction in connective tissue deposition following miR-22 inhibition was claimed, was this statistical significant? What is the definition of responders and non-responders?
- Line 355-357, missing reference(s).
- Line 387-388, “mitochondrial biogenesis HAS already shown to produce increased locomotor function and fatigue resistance”.
- 2. Animals and treatments, animal ethics approval information is missing.
- Figure 2F, the magnitude of change from week 14 onwards do not appear to be in agreement with 3H.
- Figure 2F, anti-miR-22, LNA and ASO are being used interchangeably, throughout the manuscript. need consistency. Again, fonts are too small to be read.
The manuscript is generally well-written and communicates the scientific findings clearly. However, there are multiple minor grammatical, typographical, and stylistic inconsistencies that should be addressed. These include spelling errors (e.g., “phoshorothioate” → “phosphorothioate”), incorrect word usage (e.g., “subsequentially” → “subsequently”), inconsistent abbreviation definitions, and mixed UK/US spelling conventions. A thorough copy-edit is recommended to standardize terminology, improve clarity, and enhance overall readability.
Round 2
Reviewer 1 Report
Comments and Suggestions for Authors
I thank the authors for responding to my comment. I do have an additional comment to the response that the authors provided to my first comment.
Author response: "Response 1: Thank you for your question. In this study we aim at investigating the role of miR-22 as therapeutic target for muscle wasting. Therefore, we have studied the effect of pharmacologic inhibition of miR-22 in two models of muscle wasting, one induced by environmental exposure (DIO), the second induced by genetic cause (D2-mdx). Obesity is responsible for muscle dysfunction, favoring the expansion of glycolytic fibers at the expense of low oxidative fibers, and impairing mitochondrial efficiency in energy production. Cross talk between skeletal muscles and dysfunctional adipocytes present in the enlarged visceral adipose tissue induced in obesity was confirmed to mediate muscle atrophy negatively modulating the expression of several myogenic genes such as MyoD1 and MYOG (10.2337/db14-0796). Furthermore, HFD is used to generate models of muscle atrophy in rodents (10.1016/j.archger.2023.105025, 10.3390/antiox10101633, 10.1007/s40520-015-0316-5)."
My new comment: But since you already have an atrophic mice in D2-mdx, there does not seem to be any need for inducing even more muscle atrophy through HFD. It is almost like you are elevating the miR-22 levels even more than what it already is in a D2-mdx mice only to then knock it down.
Author Response
Response from the authors to reviewers of the manuscript ijms-3859155 entitled “microRNA-22 Inhibition Stimulates Mitochondrial Homeostasis and Intracellular Degradation Pathways to Prevent Muscle Wasting” by Tomasini et al.
We thank the reviewer for taking the time to further discuss the rationale of our study. In response to the specific concern raised, we have revised the manuscript to enhance the clarity of the presentation. We hope that the updated version conveys the correct rationale of our work and meets the journal’s requirements for publication.
Comment 1: But since you already have an atrophic mice in D2-mdx, there does not seem to be any need for inducing even more muscle atrophy through HFD. It is almost like you are elevating the miR-22 levels even more than what it already is in a D2-mdx mice only to then knock it down.
Answering: "Response 1: Thank you for your question. In this study we aim at investigating the role of miR-22 as therapeutic target for muscle wasting. Therefore, we have studied the effect of pharmacologic inhibition of miR-22 in two models of muscle wasting, one induced by environmental exposure (DIO), the second induced by genetic cause (D2-mdx). Obesity is responsible for muscle dysfunction, favoring the expansion of glycolytic fibers at the expense of low oxidative fibers, and impairing mitochondrial efficiency in energy production. Cross talk between skeletal muscles and dysfunctional adipocytes present in the enlarged visceral adipose tissue induced in obesity was confirmed to mediate muscle atrophy negatively modulating the expression of several myogenic genes such as MyoD1 and MYOG (10.2337/db14-0796). Furthermore, HFD is used to generate models of muscle atrophy in rodents (10.1016/j.archger.2023.105025, 10.3390/antiox10101633, 10.1007/s40520-015-0316-5)."
Response 1: We appreciate your raising this issue, as it gives us the opportunity to improve the clarity of the experimental design presented in this study.
We would like to clarify that this study investigates a therapeutic strategy to counteract muscle atrophy exploiting two separate mouse models – a C57BL/6J DIO mouse model and a D2-mdx mouse mode – to thoroughly characterize the effect of miR-22 inhibition in muscle wasting, given the wide etiology of this condition. In our vision, employing two different models allows for a more comprehensive assessment of the molecular modulation induced by the treatment, improving confidence in the therapeutic potential of targeting this miRNA for muscle wasting conditions. Although implementing the use of HFD in a dystrophic mouse model could possibly better mimic the obese phenotype normally associated with DMD, we agree that it would also introduce several confounders that could prevent from deciphering the effect of miR-22 inhibition in muscle atrophy. Therefore, we have avoided its use in this study.
In order to improve clarity of the study design presented in our work, we have implemented several changes listed below in the Results as well as the Materials and Methods sections:
- The mouse strain was stated when each model was first presented in the Results sections as follows (changes are highlighted in bold):
- <<Next, we evaluated whether miR-22 inhibition could improve muscle health in obesity. At the highest tested dose, the treatment produced a modulatory effect on body weight in a C57BL/6 mouse model of diet-induced obesity (DIO), promoting resistance to high fat diet (HFD)-induced weight gain (Figure 2A) […]>> (lines 165-168);
- <<The myogenic effect induced by miR-22 inhibition in vivo alongside promoting a more oxidative fiber profile holds therapeutic potential to counteract muscle wasting also in chronic myopathies.>> (lines 200-202).
- <<Therefore, we investigated whether chronic pharmacologic inhibition of miR-22 could ameliorate the dystrophic muscle phenotype of a murine model of DMD (D2-mdx) over 12 weeks (Figure 3A).>> (lines 208-210).
- We stated more clearly the different nature of the two mouse models in chapter 4.3 of the Materials and Methods section as follows:
- <<Handling of animals, used diets, housing, and animal experiments performed in this study were carried out in two separate models of muscle wasting in accordance with the “Guidelines for Animal Experimentation” and under approval of “The Danish Animal Experiments Inspectorate”. >> (lines 465-468)
- << In the first mouse model mediated by diet-induced obesity (DIO),[…]>> (line 469)
- << A second model of muscle atrophy featuring muscle dystrophy was obtained purchasing 6-week-old D2.B10 (DBA/2-congenic) Dmdmdx (D2-mdx) mice […] >> (lines 478-479)
Reviewer 2 Report
Comments and Suggestions for Authors
Thank you for the point-to-point revision, I have the following minor suggestion:
- While %CFB is justified for normalization, consider including raw data in supplementary materials for transparency.
- LNA and ASO are used interchangeably in figure legends or captions.
- Ensure all figures clearly indicate statistical significance, especially where trends are discussed without significance.
- Please also include ethic application approval reference.
Author Response
Response from the authors to the reviewer of the manuscript ijms-3859155 entitled “microRNA-22 Inhibition Stimulates Mitochondrial Homeostasis and Intracellular Degradation Pathways to Prevent Muscle Wasting” by Tomasini et al.
We would like to thank the reviewer for the latest suggestions, allowing us to improve our work further. We have implemented all the requests, which can be located with the information below in the updated manuscript and supplementary materials. We hope that the revised manuscript now fulfills the requirements for publication in International Journal of Molecular Sciences.
Minor comments:
Comment 1: While %CFB is justified for normalization, consider including raw data in supplementary materials for transparency.
Response 1: Thank you for the valuable suggestion. We have now added the raw body weight data to the supplementary materials as Table 2 and specified it in lines 473, 485 and 662.
Comment 2: LNA and ASO are used interchangeably in figure legends or captions.
Response 2: Apologies for the reiterated inconsistency and thank you for noticing. We have now corrected the term “LNA” still present in the y-axis label of figure 1D and in the caption of Figure S1 with anti-miR-22 ASO. To improve consistency, Figure S1A and Figure S2C were both corrected to state CTRL.
Comment 3: Ensure all figures clearly indicate statistical significance, especially where trends are discussed without significance.
Response 3: Thank you for clarifying this problem. We have updated the figures stating the statistical significance of the data comparisons where suggested, and we updated the “statistical analysis” section 4.12 of the materials and methods accordingly.
Comment 4: Please also include ethic application approval reference.
Response 4: The information regarding the ethic application approval reference is stated in the “Institutional Review Board Statement” (line 676-679), as for editorial request. Furthermore, we have now added the information also in the materials and methods section (lines 464-467) for easier retrieval.